# Effects of Coverlys TF150^®^ on the Photosynthetic Characteristics of Grape

**DOI:** 10.3390/ijms242316659

**Published:** 2023-11-23

**Authors:** Zhonghan Li, Enshun Jiang, Minghui Liu, Qinghua Sun, Zhen Gao, Yuanpeng Du

**Affiliations:** 1College of Horticultural Science and Engineering, Shandong Agricultural University, Tai’an 271018, China; lizhh1105@163.com (Z.L.); lmhui1209@163.com (M.L.); qhsun@sdau.edu.cn (Q.S.); 2Shandong Institute of Pomology, Tai’an 271000, China; enshunjiang@163.com

**Keywords:** grape (*Vitis vinifera*), rain shelter film, photosynthesis, chlorophyll fluorescence, transcriptome, differentially expressed gene

## Abstract

Grape rain-shelter cultivation is a widely employed practice in China. At present, the most commonly used rain shelter film materials are polyvinyl chloride (PVC), polyethylene (PE), ethylene-vinyl acetate copolymer (EVA), and polyolefin (PO). Coverlys TF150^®^ is a woven fabric with an internal antifoggy PE coating that has not yet been popularized as a rain shelter film for grapes in China. To investigate the effects of Coverlys TF150^®^ on grapes, we measured the microdomain environment, leaf development, and photosynthetic characteristics of ‘Miguang’ (*Vitis vinifera* × *V. labrusca*) under rain-shelter cultivation and performed transcriptome analysis. The results showed that Coverlys TF150^®^ significantly reduced (*p* < 0.05) the light intensity, temperature, and humidity compared with PO film, increased the chlorophyll content and leaf thickness (particularly palisade tissue thickness), and increased stomatal density and stomatal opening from 10:00 to 14:00. Coverlys TF150^®^ was observed to improve the maximum efficiency of photosystem II (F_v_/F_m_), photochemical quenching (qP), the electron transfer rate (ETR), and the actual photochemical efficiency (Φ_PSII_) from 10:00 to 14:00. Moreover, the net photosynthetic rate (P_n_), intercellular CO_2_ concentration (C_i_), stomatal conductance (G_s_), and transpiration rate (T_r_) of grape leaves significantly increased (*p* < 0.05) from 10:00 to 14:00. RNA-Seq analysis of the grape leaves at 8:00, 10:00, and 12:00 revealed 1388, 1562, and 1436 differential genes at these points in time, respectively. KEGG enrichment analysis showed the occurrence of protein processing in the endoplasmic reticulum. Plant hormone signal transduction and plant-pathogen interaction were identified as the metabolic pathways with the highest differential gene expression enrichment. The psbA encoding D1 protein was significantly up-regulated in both CO10vsPO10 and CO12vsPO12, while the sHSPs family genes were significantly down-regulated in all time periods, and thus may play an important role in the maintenance of the photosystem II (PSII) activity in grape leaves under Coverlys TF150^®^. Compared with PO film, the PSI-related gene psaB was up-regulated, indicating the ability of Coverlys TF150^®^ to better maintain PSI activity. Compared with PO film, the abolic acid receptacle-associated gene PYL1 was down-regulated at all time periods under the Coverlys TF150^®^ treatment, while PP2C47 was significantly up-regulated in CO10vsPO10 and CO12vsPO12, inducing stomatal closure. The results reveal that Coverlys TF150^®^ alleviates the stress of high temperature and strong light compared with PO film, improves the photosynthetic capacity of grape leaves, and reduces the midday depression of photosynthesis.

## 1. Introduction

Vitis are one of the oldest fruit trees and are widely grown worldwide. Chinese grape production and consumption have increased rapidly in recent years; according to the statistics of the International Vine and Wine Organization (OIV) in 2020 (http://www.oiv.int/, accessed on 1 August 2022), the planting area of table grapes is 660,000 hm^2^, accounting for 84.1% of the total planting area of grapes in China, the yield ranking among the highest in the world. China has a monsoon climate, characterized by hot and rainy summers and autumns, and cold and dry winters and springs, which is disadvantageous for grape growth and facilitates the frequent occurrence of diseases in grape. Rain-shelter cultivation reduces the incidence of diseases such as downy mildew, anthrax, and powdery mildew; by covering the vine canopy with a thin film [1], the use of pesticides could be remarkably reduced. Thus, rain-shelter cultivation techniques are widely used across China, particularly in the middle and lower reaches of the Yangtze River and in the south [2]. Rain-shelter cultivation changes the microdomain environment (light intensity, temperature, and humidity) of the grape canopy. Under rain-sheltered cultivation, the photosynthetically active radiation of grape canopies is significantly reduced due to the shelter of the canopy film material [3]. Leaf morphology, anatomical structure, and physiological characteristics will be changed by changing the light intensity of the rain-shelter culture, such as leaf thickness, spongy tissue, palisade tissue, and stomatal density, thus affecting photosynthesis [4,5].

Since the 20th century, the global climate has warmed, particularly during fruit ripening, with extreme temperatures reaching 45 °C and lasting up to two weeks in some years [6]. High temperature stress mainly damages the structure and function of biofilms by producing excessive reactive oxygen species [7]. The photochemical reaction of PSII is highly sensitive to extreme temperatures, and its activity will decrease or even be partially inactivated at high temperatures [8]. High temperatures also reduce the overall activity of PSII by inhibiting the maximum light repair rate of plants. Plants activate defense mechanisms after exposure to high temperature stress, and the expression of many heat shock protein genes can change significantly in a very short period of time. This is particularly true for heat sensitive factors (HSFs) and heat shock proteins (HSPs) [9], which are involved in the photochemical reactions that protect PSII. Light intensity typically exceeds 1800 μmol m^−2^·s^−1^ under natural conditions [10], which leads to excess incoming light energy that can damage the carotene, chlorophyll, and D1 proteins if not dissipated or utilized in time, thereby inhibiting PSII [11]. High temperatures are often accompanied by intense light, which causes compound damage to plants. The synergistic stress of high temperatures and intense light causes greater damage to plants than that of high temperatures or intense light alone [12]. High temperatures and strong light stress reduce the repair ability of plant PSII through the production of ROS [13].

Different film materials have certain effects on light and temperature. At present, the main film materials used for rain shelters are polyvinyl chloride (PVC), polyethylene (PE), ethylene-vinyl acetate copolymer (EVA), and polyolefin (PO) film [14,15]. Studies have shown that compared with open field cultivation, the net photosynthetic rate (P_n_) of leaves covered by PVC film are significantly reduced, while no significant reductions are observed in the maximum efficiency of photosystem II (F_v_/F_m_), indicating that rain-shelter cultivation does not damage the photosystem II (PSII) response center [5]. Tian et al. (2019) found that the daily variation of the photosynthetic rate (P_n_) of leaves covered by PE film exhibited unimodal characteristics, which alleviated midday light stress and improved the ability of Chinese cherry leaves to utilize weak light [16]. Compared with shelter-free, the total daily photosynthetic accumulation of leaves under rain-shelter cultivation increases; this is because plants can successfully adapt to the low-par conditions under the rain shelters, avoiding the negative effects of midday strong light on photosynthesis. Meng et al. (2013) revealed an increase in grape canopy temperature compared to the control by covering with PO film [2]. Coverlys TF150^®^ is a single-coated PE fabric that protects fruit against extreme climate conditions. The application results of Coverlys TF150^®^ in Southern Italy showed that compared with PE film, the photosynthetically active radiation was reduced, but the ultraviolet radiation (280~380 nm) was increased by 127%, which reduced the negative impact of strong light stress on the top leaves, and thus improved the photosynthetic efficiency [17]. In this study, the microdomain environment, leaf development, and photosynthetic characteristics and transcriptome of grape under rain-shelter cultivation with Coverlys TF150^®^ were analyzed, which provides a theoretical basis for its application in continental monsoon climate regions.

## 2. Results

### 2.1. Effects of Coverlys TF150^®^ on the Grape Canopy Microdomain Environment

The photosynthetically active radiation of the Coverlys TF150^®^ grape canopy was lower than that of PO film (Figure 1A,B). The average transmittance of Coverlys TF150^®^ and PO film was 65.23% and 74.05%, respectively. The photosynthetically active radiation of PO film in the field reached 2073 μmol·m^−2^·s^−1^, while that of Coverlys TF150^®^ was only 1703 μmol·m^−2^·s^−1^. The PO film photosynthetically active radiation over 1600 μmol·m^−2^·s^−1^ measured at 12:00 accounted for 32.14% of the total, while that of Coverlys TF150^®^ accounted for 5.95%. Comparing the diurnal variation of photosynthetically active radiation of the two canopy film materials on sunny days reveals that the light intensity of both materials was similar before 8:00, exceeding 800 μmol·m^−2^·s^−1^. There was a significant difference in photosynthetically active radiation between 10:00 and 14:00 (Figure 1B). The Coverlys TF150^®^ 1400–1600 μmol·m^−2^·s^−1^ light intensity measured at 12:00 accounted for 22.62% of the total, while that of PO accounted for 14.29%. The light intensity of Coverlys TF150^®^ at 12:00 (800–1400 μmol·m^−2^·s^−1^) accounted for 57.14%, while that of PO film was only 33.33%.

Coverlys TF150^®^ reduced the temperature (Figure 1C,D) and humidity (Figure 1E,F) of the grape canopy compared to PO film. The maximum temperature of the grape canopy under Coverlys TF150^®^ was 40.80 °C, which was 2.00 °C lower than that under PO film. The minimum humidity of the grape canopy under Coverlys TF150^®^ was 1.00% lower than that under PO film. Under Coverlys TF150^®^, the proportion of humidity ≥80% in the grape canopy was 22.07%, which was 15.86% lower than that under PO film, while the proportion of humidity between 40% and 60% was 10.68% higher than that under PO film.

### 2.2. Effects of Coverlys TF150^®^ on the Grape Canopy Microdomain Environment

Compared with PO film, Coverlys TF150^®^ significantly increased the thickness, mass, and chlorophyll content of grape leaves (Figure 2A–D). For example, the leaf thickness, weight, and chlorophyll content increased by 23.53%, 13.32%, and 36.36%, respectively, but no significant differences were observed in the leaf area between the two mulching methods. The palisade tissue of grape leaves under Coverlys TF150^®^ was significantly higher than that under PO film (Figure 2G; Table 1), increasing by 13.26% compared with PO film. Moreover, the epidermal thickness and spongy tissue thickness of grape leaves under Coverlys TF150^®^ were higher than those of PO film, while the lower epidermis thickness was lower than that of PO film, yet the difference was not significant. The results suggest that the Coverlys TF150^®^ shed film material increases the thickness of the leaves by thickening the fence tissue.

Grape leaves under Coverlys TF150^®^ and PO film were scanned by electron microscope every two hours from 8:00 to 16:00 on a sunny day. The results showed that most stomata openings of grape leaves under Coverlys TF150^®^ were greater than those under PO film from 10:00 to 14:00, and the opposite was true at 8:00 and 16:00 (Figure 2F–I). The stomatal density of grape leaves under Coverlys TF150^®^ was also observed to be significantly higher than that of PO film, increasing by 14.93% compared with PO film (Figure 2E). The results reveal that Coverlys TF150^®^ facilitates the opening of stomata in grape leaves between 10:00 and 14:00.

### 2.3. Effects of Coverlys TF150^®^ on Photosynthetic Characteristics of Grape Leaves

The net photosynthetic rate of grape leaves under PO film at 8:00 was higher than that of Coverlys TF150^®^. However, during 10:00–14:00, the net photosynthetic rate of grape leaves under Coverlys TF150^®^ was significantly higher than that under PO film (Figure 3A). Moreover, the net photosynthetic rate area of grape leaves under Coverlys TF150^®^ was determined to be 365,520 μmol·m^−2^, 8.75% higher than PO film. Compared with PO film, the intercellular CO_2_ concentration (C_i_) of grape leaves decreased significantly under Coverlys TF150^®^ at 8:00, while that between 10:00 and 14:00 was lower than that of PO film, with no significant difference at 16:00 (Figure 3B). In addition, the trends in stomatal conductance (G_s_) and transpiration rate (T_r_) were similar to those of the net photosynthetic rate. At 10:00–14:00, the stomatal conductance (G_s_) and transpiration rate (T_r_) of grape leaves under Coverlys TF150^®^ were significantly higher than those under PO film (Figure 3C,D). These results indicate that the change in net photosynthetic rate is caused by stomatal factors.

Under Coverlys TF150^®^, F_v_/F_m_ and Φ_PSII_ of grape leaves reached their highest point of the day at 10:00 (Figure 3E,F). the daily changes of grape leaves ETR and qP followed a bimodal trend, which appeared at 10:00 and 14:00, respectively (Figure 3G,H). The values of grape leaf F_v_/F_m_, Φ_PSII_, ETR, and qP under Coverlys TF150^®^ were significantly higher than PO film at 10:00–14:00, yet the opposite was true at 8:00 and 16:00. The results reveal the ability of Coverlys TF150^®^ to reduce the stress degree of grape leaves at 10:00–14:00 and increase PSII activity.

### 2.4. Analysis of Differentially Expressed Genes

We adopted the screening conditions |log_2_FC| > 0.58 and *p*-value < 0.05 to identify the differentially expressed genes. There were 1388 differential genes in CO8vsPO8, including 740 up-regulated genes and 646 down-regulated genes (Figure 4A). A total of 1562 differential genes were detected in CO10vsPO10, including 796 up-regulated genes and 766 down-regulated genes (Figure 4B). A total of 1436 differential genes were detected in CO12vsPO12, including 822 up-regulated genes and 614 down-regulated genes (Figure 4C). In addition, 463 differential genes were shared by CO8vsPO8 and CO10vsPO10, while 458 differential genes were shared by CO8vsPO8 and CO12vsPO12. CO10vsPO10 and CO12vsPO12 shared 462 differential genes, while Coverlys TF150^®^ and PO film shared 232 differential genes at 8:00, 10:00, and 12:00 (Figure 4D).

### 2.5. KEGG Enrichment Analysis of Differential Genes Expression

KEGG pathway analysis was performed on the differentially expressed genes to demonstrate their corresponding metabolic processes and signaling pathways more intuitively (Figure 5). The results reveal that protein processing in endoplasmic reticulum, plant hormone signal transduction, plant-pathogen interaction, flavonoid biosynthesis, circadian rhythm-plant, and MAPK signaling pathway-plant were enriched at 8:00, 10:00, and 12:00.

### 2.6. KEGG Enrichment Analysis of Differential Genes

In the comparison of CO8 and PO8, the photosynthesis genes were mostly down-regulated, while the comparison of the CO10vsPO10 and CO12vsPO12 groups revealed photosynthetic-related genes to be up-regulated (Figure 6). The expression level of the psbA encoding D1 protein was not significantly different in CO8vsPO8, yet it was significantly up-regulated in both CO10vsPO10 and CO12vsPO12. A single psaB gene was down-regulated in CO8vsPO8, two psaB genes were up-regulated in CO10vsPO10, and one psaB gene was up-regulated in CO12vsPO12. Moreover, psbC, psbD, psbH, atpA, and atpB were all significantly down-regulated in CO8vsPO8, while PsaL, psbK, petC, and atpL were significantly up-regulated in CO12vsPO12.

### 2.7. Comparison of ABA Pathway Gene Expression in Grape Leaves under Coverlys TF150^®^ and PO Film

In this study, a total of 18 differential genes were associated with the abscisic acid (ABA) pathway genes (Figure 7). The majority of ABA receptor-related genes of Coverlys TF150^®^ were down-regulated compared to PO film. PYL1 was observed to be significantly down-regulated in the CO8vsPO8, CO10vsPO10, and CO12vsPO12 comparisons. The two PYL4 exhibited a significant down-regulation in the comparison of CO10 and PO10, while PYL8 was significantly down-regulated in CO12vsPO12. PYL11 was significantly up-regulated in CO10vsPO10 and CO12vsPO12, while PP2C47 was significantly up-regulated in CO10vsPO10 and CO12vsPO12. PP2C44 and PP2C63 were significantly up-regulated in CO8vsPO8 and CO10vsPO10. PP2C40 and PP2C27 were significantly down-regulated in CO8vsPO8, while PP2C14 was significantly up-regulated. PP2C23 and PP2C58 were significantly up-regulated in CO10vsPO10. In addition, PP2C9 was significantly up-regulated in CO12vsPO12, while SAPK3 was significantly up-regulated in CO8vsPO8 and CO12vsPO12. SAPK7 was significantly up-regulated in CO10vsPO10, and SAPK2 was significantly down-regulated in CO8vsPO8. Moreover, ABF3 was significantly up-regulated in CO8vsPO8 and significantly up-regulated in CO12vsPO12.

### 2.8. Comparison of Heat Shock Protein and Heat Shock Protein Transcription Factor Gene Expression in Grape Leaves under Coverlys TF150^®^ and PO Film 

HSPs are a group of proteins produced by cells under stress, and HSPs-related genes are derived from the “endoplasmic reticulum protein processing” pathway. The comparison of C08vsPO8, C010vsPO10, and C012vsPO12 revealed 24 HSPs to be DEGs, most of which were plant small heat shock proteins (sHSPs), and the remaining were Hsp70 and Hsp90 families (Figure 8). The expression levels of the majority of heat shock protein-related genes under the Coverlys TF150^®^ treatment was lower than that under the PO film treatment, and the expression level was most obvious at 8:00. The sHSPs family genes were significantly down-regulated in all time periods, and HSP22.0 and HSP17.3-D were the most significantly down-regulated. With the exception of two HSP70 genes that were not significantly different in C010vsPO10, the majority of the HSP70 family genes were down-regulated. In addition, the Hsp90 family genes also exhibited a downward trend in several time periods.

HSFs are key transcription factors in the high temperature response of plants. Two HSF30 were identified to be significantly down-regulated in the comparison between CO8 and PO8, while one HSF30 was significantly down-regulated in the comparison of CO10 and PO10. Furthermore, HSFB2A, HSFA2D, and HSF24 were significantly down-regulated in the comparison of CO8 and PO8, while HSF24 was significantly down-regulated in CO10vsPO10 (Figure 8).

## 3. Discussion

### 3.1. Effects of Coverlys TF150^®^ on Leaf Morphology, Anatomical Structure, and Stomata of Grape

The leaf is the most sensitive organ to environmental changes amongst all plant vegetative organs. It is also the core organ for photosynthesis processes, which are strongly affected by environmental conditions such as temperature and light [18]. Our results reveal that Coverlys TF150^®^ can reduce light intensity and temperature and improve the microdomain environment compared with PO film during the growing season. Previous studies have shown that high temperature and strong light stress can also affect leaf morphology and structure, resulting in a reduced thickness of the epidermis, palisade tissue, and spongy tissue [19]. Furthermore, high temperatures and strong light stress reduce the thickness of grape leaves under PO film, particularly the palisade tissue thickness. High temperatures [20] and strong light [21] stress have also been reported to significantly reduce the chlorophyll a and b contents of leaves, resulting in a lower chlorophyll content of grape leaves under PO film compared to Coverlys TF150^®^.

Stomates act as channels for gas exchange and water loss between the internal and external environment of a plant. High temperatures [22] and strong light stress [23] will reduce stomatal density and induce stomatal closure. In our experiment, the pores of grape leaves under PO film were severely stressed by high temperatures and strong light from 10:00, and thus the stomata were closed in order to retain water. ABA plays an important role in regulating plant response to stress. It is considered as a key signaling molecule that accumulates under biological and abiotic stress [24], and is a hormone that aids plants to alleviate abiotic stress. High temperature stress may affect its biosynthesis and signal transduction processes [25]. When stress occurs, ABA will accumulate rapidly and in large quantities. Under abiotic stress, the concentration of ABA in plants increases and it binds to the ABA receptor PYR/PYL/RCAR. PYR/PYL/RCAR is a protein regulator of PP2C activity that initiates abscisic acid signal transduction by inhibiting PP2C phosphatase activity [26,27]. It then activates transcription factors through the phosphorylation of SnRK2s, promotes the expression of ABA response genes, and activates SLAC1 and KAT1 ion channels, thereby reducing the pressure of defense cells by ion efflux and inducing stomatal closure [28,29]. Compared with PO film, at 10:00 and 12:00, the majority of ABA receptors-related genes in grape leaves were significantly down-regulated, while PP2C47 was significantly up-regulated in CO10vsPO10 and CO12vsPO12. This indicates that Coverlys TF150^®^ did not remove the inhibition of PP2C on SnRK2s and reduced stomatal closure (Figure 7).

### 3.2. Effects of Coverlys TF150^®^ on Chlorophyll Fluorescence and HSPs in Grape Leaves

Fluorescence parameters are probes used to monitor plant resistance to stress, with the ability to reflect the degree of plant damage and help explain photosynthesis. PSII is the site of initial photoinhibition [30,31]. The F_v_/F_m_ value of plant leaves is approximately 0.8 under normal circumstances, and typically decreases when plants are stressed. In this study, we observed the F_v_/F_m_ value of grape leaves under Coverlys TF150^®^ to be significantly higher than that of PO film from 10:00 to 14:00. Moreover, although the F_v_/F_m_ value of grape leaves under Coverlys TF150^®^ decreased at 12:00, it was not lower than 0.8. The results indicate that Coverlys TF150^®^ alleviates the stress of high temperatures and strong light, which is conducive to the progress of photosynthesis. Φ_PSII_, ETR, and qP exhibited similar change trends. From 10:00 to 14:00, the Φ_PSII_, ETR, and qP of grape leaves under Coverlys TF150^®^ were significantly higher than those of PO film. The reduced Φ_PSII_, ETR, and qP indicated that the PSII was damaged, the photoreaction was inhibited, the photochemical efficiency was reduced, and electron transport was hindered. This indicates the ability of Coverlys TF150^®^ to reduce the damage caused by high temperatures and strong light to PSII from 10:00 to 14:00, allowing the leaf to maintain a high photochemical activity and stabilize the electron transfer.

Among the many protein subunits of PSII, the D1 protein in the reaction center is the most vulnerable to high temperatures (35 °C) [32] and strong light (≥1500 μmol·m^−2^·s^−1^) [33] attack. When exposed to ROS, the D1 protein is easily degraded by the corresponding protease [34] and can be rapidly degraded and resynthesized continuously to maintain the PSII activity [35]. The D1 protein-related gene psbA was observed to be significantly up-regulated in both the CO10vsPO10 and CO12vsPO12 comparisons, suggesting that Coverlys TF150^®^ alleviates heat and light stress by resynthesizing the D1 protein. Strong light not only disrupts the transcription and post-transcriptional translation of the plant photosystem gene psbA, but also reduces intracellular ATP content. In addition, the substrate ATP level is correlated with the D1 protein biosynthesis rate [36]. In our analysis, the ATP-related gene atpL was also up-regulated in the CO12-PO12 comparison, suggesting that Coverlys TF150^®^ grape leaves enhance the rate of D1 protein resynthesis by increasing ATP levels. Photosystem I (PSI) is a pigment protein complex composed of multiple protein subunits integrated on the photosynthetic film. In the protein subunits constituting PSI, PsaA and PsaB are bound in a heterodimer state as the reaction center. PSI is relatively stable and generally not damaged. When the number of electrons flowing to PSI through PSII exceeds the number acceptable by the electron acceptor on the reducing side of PSI, strong light will damage PSI [37]. Under Coverlys TF150^®^, two psaB genes from the PSI-related genes in grape leaves were up-regulated in CO10vsPO10, while one psaB gene was up-regulated in CO12vsPO12. In summary, the PSI activity of grape leaves under PO film was inhibited.

### 3.3. Effects of Coverlys TF150^®^ on Chlorophyll Fluorescence and HSPs in Grape Leaves

Plant photosynthesis is mainly affected by environmental conditions, and the optimal temperature for grape photosynthesis is 25–35 °C [38]. When the temperature exceeds 37 °C, both the PSII reaction center and the electron acceptor side are damaged, resulting in serious photoinhibition, which consequently affects the photochemical reaction on the chloroplast thylakoid film and carbon fixation process in the matrix [39]. Moreover, PSII activity decreases or is even partially inactivated [8], which also results in the degradation of the ribulose 1, 5-diphosphate carboxyl enzyme (Rubisco) [40]. The degradation of the Rubisco binding protein (RBP) and Rubisco small subunits [41] inhibit Rubisco activating enzyme, and ultimately lead to a decrease in photosynthetic efficiency [42] and photosynthetic products. Table grapes in the southern region of China often experience extreme high temperatures, which cause serious damage to their leaves and fruits. In 2022, Hunan, Hubei, Jiangsu, Fujian, and other southern regions continuously experienced high temperatures above 40 °C, and the Kyoho grape producing area in Fu’an of Fujian province had been experiencing high temperatures of over 40 °C for nearly 50 days. Grapes have experienced symptoms such as leaf burns, fruit burns, fruit shrinkage, and softening. In our study, compared with PO film, Coverlys TF150^®^ reduced the canopy temperature, and the maximum temperature was 2 °C lower than that of PO film. This is conducive to alleviating the damage caused by high temperature stress on plants and promoting the progress of photosynthesis.

Light is another key factor affecting photosynthesis, and the net photosynthetic rate changes with the light intensity throughout the day. Insufficient light will cause a shortage of assimilatory power, and thus the key photosynthetic enzymes are not fully activated and restrict photosynthesis. The net photosynthetic rate of grape leaves under Coverlys TF150^®^ was significantly lower than that of PO film at 8:00 and 16:00. This can be attributed to the lower effective photosynthetic radiation at 8:00 and 16:00, with the light intensity of Coverlys TF150^®^ exceeding that of PO film, which was lower than the suitable photosynthesis range. However, too high a light intensity will cause plants to absorb more light energy than they can use, resulting in the accumulation of excess excitation energy [7]. This generates ROS and causes serious light suppression. The diurnal variation of grape leaf photosynthesis under Coverlys TF150^®^ exhibited a bimodal trend, and the net photosynthetic rate peaked at 10:00 and 14:00. In addition, at this time period, the net photosynthetic rate of grape leaf under Coverlys TF150^®^ was significantly higher than that under PO film. Since Coverlys TF150^®^ alleviates high temperature and strong light stress from 10:00 to 14:00, the photosynthesizing time is improved. The variation trend of stomatal conductance is similar to that of the net photosynthetic rate, and thus it can be seen that the change in the net photosynthetic rate is caused by stomatal factors. Under sunny conditions in the field, light suppression occurred at 9:00 in grape leaves, and the outside temperature was higher from 10:00 to 14:00, accompanied by strong light. The light saturation point of most grape varieties was approximately 1400 μmol·m^−2^·s^−1^. It was found that under 1600 μmol·m^−2^·s^−1^ strong light, the PSII reaction center and electron acceptor side were significantly inhibited in the suitable temperature range of 28–34 °C, while the light transmittal rate of PO film was higher, and the maximum light intensity reached 2073 μmol·m^−2^·s^−1^. The high temperature and strong light stress were serious, causing the photosynthetic “nap” phenomenon (the photosynthetic rate of plants decreases at noon) of grape leaves and reducing the net photosynthetic rate. The net photosynthetic rate of grape leaves under Coverlys TF150^®^ was maximized at 10:00. In particular, the lower temperature at 10:00 results in a higher light intensity at this time than that at 8:00, yet it does not exceed the compensation point of grape leaf photosynthesis. Although the temperature at 14:00 is slightly higher than at 12:00, the light at 14:00 is lower, resulting in a second net photosynthetic rate peak at 14:00. Moreover, the daily variation area of the Coverlys TF150^®^ net photosynthetic rate was 8.75% higher than that of PO film. This indicates that Coverlys TF150^®^ was beneficial to the photosynthesis of grape leaves, accelerating the daily photosynthetic accumulation of plants.

## 4. Materials and Methods

### 4.1. Test Site and Materials

The experiment was conducted in 2021 at the vineyard of the Horticultural Experiment Station of Shandong Agricultural University (117°16′ E, 36°16′ N), Tai’an, China. Tai’an has a warm and temperate semi-humid monsoon climate. The annual average temperature, precipitation, and sunshine duration are 13 °C, 700–800 mm, and 2627 h, respectively, and the frost-free period is more than 200 d (http://data.cma.cn, accessed on 1 August 2022).

The 4-year-old grape variety ‘Miguang’ (‘Kyoho’ × ‘Zaoheibao’) was selected as the test material. The polythene fabric Coverlys TF150^®^ directly covered the trunk over the top of the vine. Both sides of the fabric were bound to drawing wire with cloth strips through preset holes. The PO film was used with a simple steel frame rain shelter (75 m length and 2.5 m width). The highest point of the mulch was 0.7 m above the tree body. The plots were grown under a trellis inverted L-shaped horizontal canopy with 1.5 m plant spacing and 3 m row spacing. The PO film mulch and Coverlys TF150^®^ mulch treatments were set up with 3 replicates per treatment and 10 vines per replicate using a random block arrangement.

### 4.2. Microdomain Environmental Monitoring

Photosynthetically active radiation was measured using three sets of photosynthetically active radiation monitors WatchDog 2000 (SPECTRUM, Madison, WI, USA) installed in the same area, approximately 5 cm above the grape canopy covered by Coverlys TF150^®^ and PO film. Three sets of temperature and humidity detectors L92-1 (LUGE, Hangzhou, China) were installed to detect the temperature and humidity of the fruit surface in real time, approximately 5 cm above the grape canopy covered by Coverlys TF150^®^ and PO film. Temperature and humidity were recorded every 0.5 h.

### 4.3. Determination of Grape Leaf Area, Leaf Thickness, Leaf Weight, and Chlorophyll Content

#### 4.3.1. Determination of Grape Leaf Area, Leaf Thickness, and Leaf Weight

A total of 30 leaves at the 9th node were collected in mid-July 2021 and placed on coordinate paper. Photographs were then taken of the leaves and the leaf area was estimated using Digimizer (v6.0.0). The leaf weight was measured with a balance (0.01 g accuracy) and the leaf thickness was determined with vernier calipers (0.01 mm accuracy).

#### 4.3.2. Determination of Chlorophyll Content in Grape Leaves

Leaf chlorophyll content was determined by the acetone extraction colorimetric method. An amount of 30 leaves at the 9th node were collected for each treatment. A total of 0.2 g of the leaf samples were placed in 25 mL test tubes with 10 mL of 80% acetone until completely faded, the absorbance was measured at 470, 646, and 663 nm using a TU-1900 ultraviolet spectrophotometer (PERSEE, Beijing, China), and the chlorophyll content was subsequently calculated. Biological repetition was performed three times for each treatment. The contents of chlorophyll a (Chla), chlorophyll b (Chlb), and total chlorophyll (Chl <a+b>) were calculated using the equations developed by Tian et al. (2019) [16]:Chla = 12.21A663 − 2.81A646Chlb = 20.13A646 − 5.03A663Chl(a + b) = Chla + Chlb

### 4.4. Preparation and Observation of Paraffin Sections of Grape Leaves

In mid-July, 30 leaves at the 9th node were collected and 1 mm^2^ leaf squares were removed from each leaf vein. The samples were placed in formalin-acetic acid-alcohol (FAA) fixing solution and sliced [43]. The slices were then moved in environmental dewaxing clear solution I and environmental dewaxing clear solution II for 20 min successively, and then placed in anhydrous ethanol I, anhydrous ethanol II, and 75% alcohol for 5 min successively. Following this, the slices were rinsed with running water, dyed in the plant coloring solution for 2 h, and washed lightly with tap water until the excess dye was removed. The slices were then placed into 50%, 70%, and 80% gradient alcohol for 3–8 s successively, then into the plant solid green dyeing solution for 6–20 s, and dehydrated in three tanks of anhydrous ethanol. Following this, the slices were placed into clean xylene for 5 min. Finally, the slices were sealed with neutral gum and images were collected and analyzed using a microscope. Biological repeats were performed five times for each leaf index.

### 4.5. Determination of the Photosynthetic Characteristics of Grape

#### 4.5.1. Determination of Grape Leaf Photosynthesis

On a sunny day in the middle of July, the leaves at the 9th nodes of the mean branch on the sunny side of the plant were randomly selected under the two types of rain shelter film and measured every two hours from 8:00 with a Ciras-3 photosynthesis system (PP System, Amesbury, MA, USA). The net photosynthetic rate (P_n_), stomatal conductance (G_s_), intercellular CO_2_ concentration (C_i_), and transpiration rate (T_r_), amongst other indicators, were recorded. Measurements were repeated 15 times. The leaf cuvette of the photosynthesis system was square, with dimensions of 25 mm × 18 mm and a light intensity set to 1200 μmol·m^−2^·s^−1^. During measurements, the leaf was gripped with the leaf cuvette such that it filled the leaf cuvette. Once the indicator number stabilized, the reading was recorded.

#### 4.5.2. Determination of Chlorophyll Fluorescence in Grape Leaves

Following the method of Kooten and Snell (1990) [44], the diurnal variation of the chlorophyll fluorescence parameters of grape leaves at the 9th nodes were measured between 8:00 and 16:00 using a portable pulse-modulated FMS-2 fluorometer (Hansatech, Inc., Pentney, UK) on a sunny day in mid-July. A total of nine fully expanded leaves were used for investigation. The measurement procedure is described as follows. Under light adaptation (activation light intensity of approximately 1000 μmol·m^−2^·s^−1^), the leaves of each strain were first exposed to 60 s of action light (1000 μmol·m^−2^·s^−1^), and then to a measuring light (<0.05 μmol·m^−2^·s^−1^) to measure the minimum fluorescence (F_o_′) of the leaves. The maximum fluorescence value (F_m_′) under light adaptation was measured by a saturated pulse light (12,000 μmol·m^−2^·s^−1^). Steady-state fluorescence (F_s_) was obtained after Ft stabilization under the applied light. The initial fluorescence (F_o_) was determined after dark adaptation for 30 min by closing the blade clamp. The maximum fluorescence (F_m_) under dark adaptation was measured using a 12,000 μmol·m^−2^·s^−1^ saturated pulsed light to reduce the primary electron acceptor QA.

### 4.6. RNA Extraction, cDNA Library Preparation, and Sequencing

RNAiso Plus (TaKaRa, Beijing, China) was used to extract total RNA from the mixed sample according to the manufacturer’s instructions. RNA purity and quantification were determined using the NanoDrop 2000 spectrophotometer (Thermo Scientific, Waltham, MA, USA). RNA integrity was assessed using the Agilent 2100 Bioanalyzer (Agilent Technologies, Santa Clara, CA, USA). The VAHTS Universal V6 RNA-seq Library Prep kit was used to construct transcriptome libraries according to the manufacturer’s instructions and index codes were added to the attribute sequence of each sample. In brief, the mRNA was isolated with a magnetic bead, which poly-T oligo-attached, and the cDNA was then synthesized. The QIA Quick PCR kit was used to purify the library fragments, and EB buffer was used for the end repair, A-tail, and adapter addition. For the Quality Control (QC) step, the sample libraries were quantified and identified using the Agilent Bioanalyzer 2100 system (Agilent Technologies, CA, USA) and the StepOne Plus™ Real-Time PCR system (library effective concentration > 10 nM). Finally, the lllumina Novaseq 6000 sequencing platform was used to sequence the library and 150 bp double-ended reads were generated. Grape leaves of the 9th section were selected at 8:00, 10:00, and 12:00, respectively, on a sunny day in the middle of July and stored at −80 °C after quick-freezing with liquid nitrogen. Three biological replicates were used in the RNA-Seq experiment to construct the library, with a total of 18 samples. Based on the hypergeometric distribution, GO, KEGG pathway, Reactome and WikiPathways enrichment analysis of DEGs were performed to screen the significant enriched term using R (v3.2.0), respectively. R (v3.2.0) was used to draw the column diagram, the chord diagram, and bubble diagram of the significant enrichment term. Gene Set Enrichment Analysis (GSEA) was performed using GSEA software (v4.3.2 for Windows). The analysis used a predefined gene set, and the genes were ranked according to the degree of differential expression in the two types of samples. Then whether the predefined gene set was enriched at the top or bottom of the ranking list is tested.

### 4.7. Data Statistics

Microsoft Excel 2016 (Microsoft Corp, Armonk, NY, USA) was used for data processing and mapping, SPSS 26.0 (SPSS, Chicago, IL, USA) was adopted for independent sample *t*-test analysis, and the least significant difference (LSD) method was employed to test for significant differences.

## 5. Conclusions

In continental monsoon climate regions, compared with PO film, Coverlys TF150^®^ alleviates stomatal closure of leaves at 10:00 and 14:00 by regulating ABA-related gene expression, down-regulates heat shock protein and alleviates the effect of strong light on photosystem. As a consequence, it improves the leaf quality, thickness, and chlorophyll content, alleviates the midday depression of photosynthesis, and improves photosynthetic capacity. Therefore, Coverlys TF150^®^ material has good application prospects in continental monsoon climate regions.

## Figures and Tables

**Figure 1 ijms-24-16659-f001:**
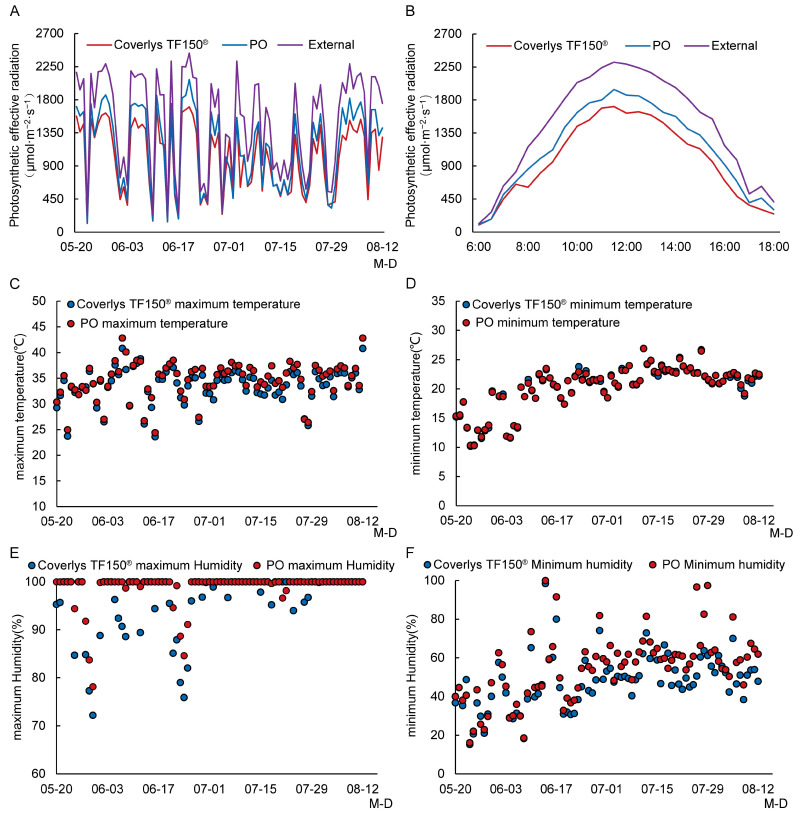
Effects of Coverlys TF150^®^ and PO film on the grape canopy microdomain environment. (**A**) Photosynthetically active radiation measured at 12:00. (**B**) Photosynthetically active radiation measured at 12:00. (**C**) Maximum temperature. (**D**) Minimum temperature. (**E**) Maximum humidity. (**F**) Minimum humidity. Note: M-D stands for date, M stands for Month, D stands for day.

**Figure 2 ijms-24-16659-f002:**
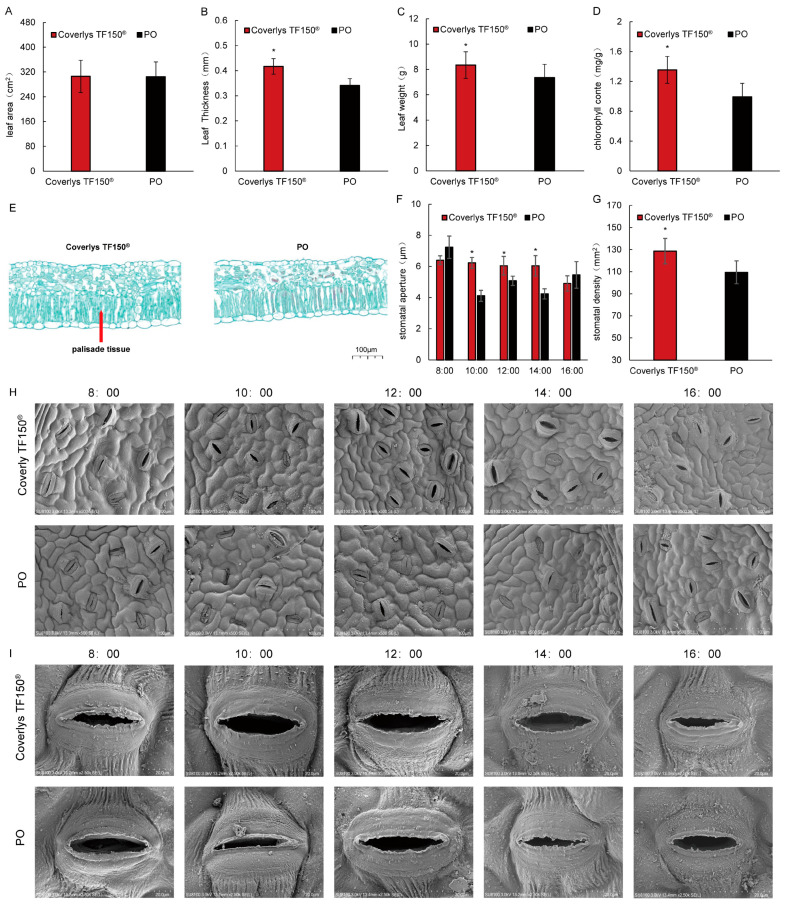
Effects of Coverlys TF150^®^ and PO film on the growth and development of grape leaves. (**A**) Leaf area. (**B**) Leaf thickness. (**C**) Leaf weight. (**D**) Chlorophyll content. (**E**) Comparison of Coverlys TF150^®^ and PO film paraffin sections. (**F**) Stomatal opening. (**G**) Stomatal density. (**H**,**I**) Stomatal phenotype (Scale bar: 100 μm and 20 μm). Note: * indicates significant differences (*p* < 0.05).

**Figure 3 ijms-24-16659-f003:**
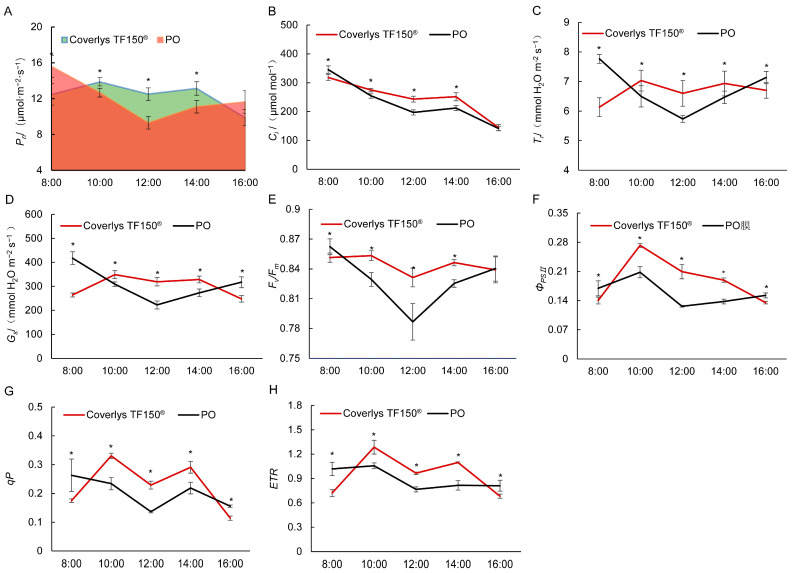
Effects of Coverlys TF150^®^ and PO film on the photosynthetic characteristics of grape leaves. (**A**) Net photosynthetic rate, P_n_. (**B**) Stomatal conductance, G_s_. (**C**) Intercellular CO_2_ concentration, C_i_. (**D**) Transpiration rate, T_r_. (**E**) Maximum PSII quantum yield, Fv/Fm. (**F**) Actual photochemical efficiency of PSII, Φ_PSII_. (**G**) Photochemical quenching coefficient, qP. (**H**) Electron transport rate, ETR. Note: * indicates significant differences (*p* < 0.05).

**Figure 4 ijms-24-16659-f004:**
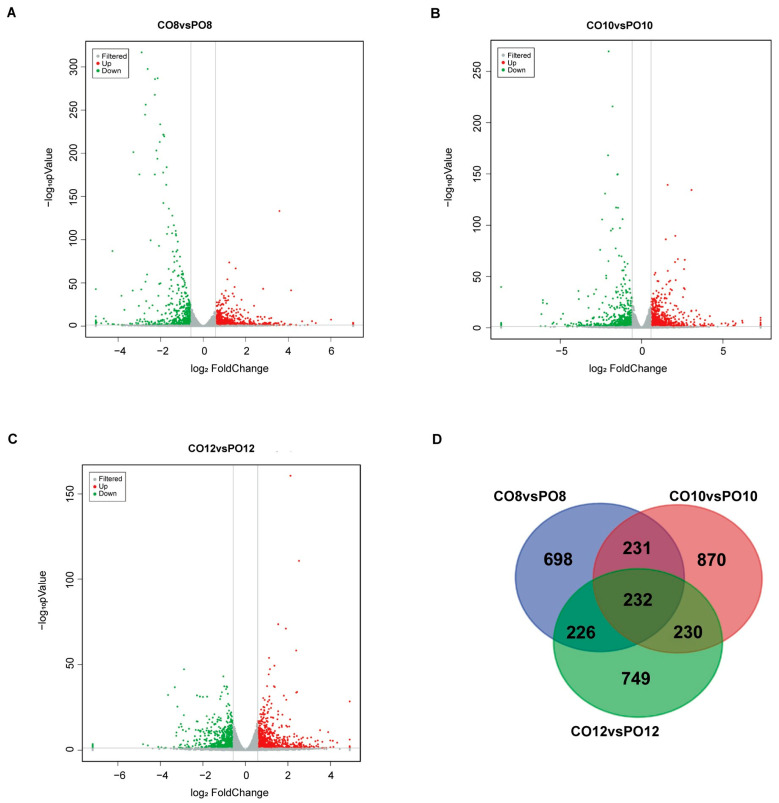
Analysis of gene differential expression in leaves of Coverlys TF150^®^ and PO film at 8:00, 10:00, and 12:00. (**A**–**C**) CO8vsPO8, CO10vsPO10, and CO12vsPO12 volcano map analysis. log_2_(fold change) transformation was used for the x-axis and log_10_(*p* value) was used for the y-axis. Red represents up-regulation, green represents down-regulation, and gray represents no significant difference between the experimental and control groups. (**D**) Venn diagram showing the number of differentially expressed genes in grape leaves under Coverlys TF150^®^ and PO film at 8:00, 10:00, and 12:00. note: CO stands for Coverlys TF150^®^, PO stands for PO film, 8, 10, and 12 represent 8:00, 10:00, and 12:00, respectively.

**Figure 5 ijms-24-16659-f005:**
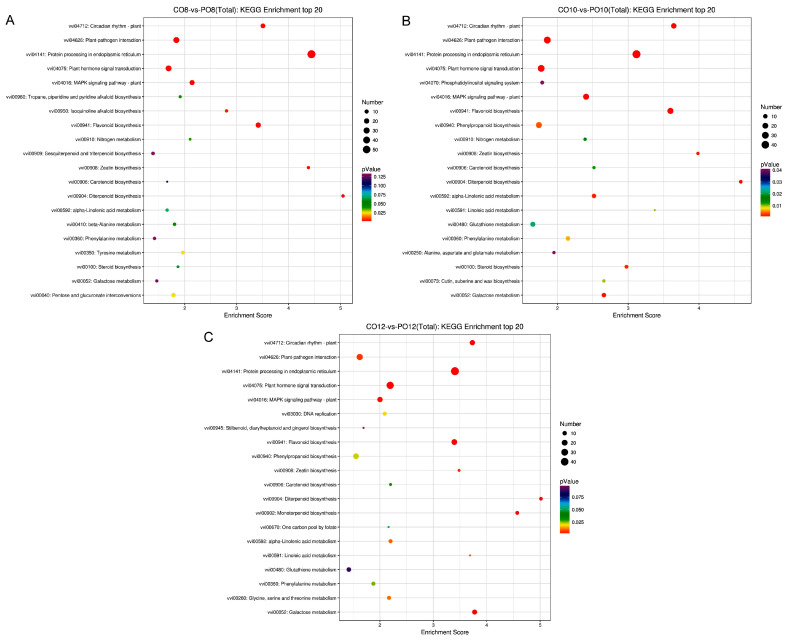
Bubble diagram of the top 20 KEGG enrichment pathways. Entries with larger bubbles contain more difference-protein coding genes. The smaller the enrichment *p*-value, the greater the significance. (**A**–**C**) CO8vsPO8, CO10vsPO10, and CO12vsPO12 KEGG enriched top 20 bubble diagram.

**Figure 6 ijms-24-16659-f006:**
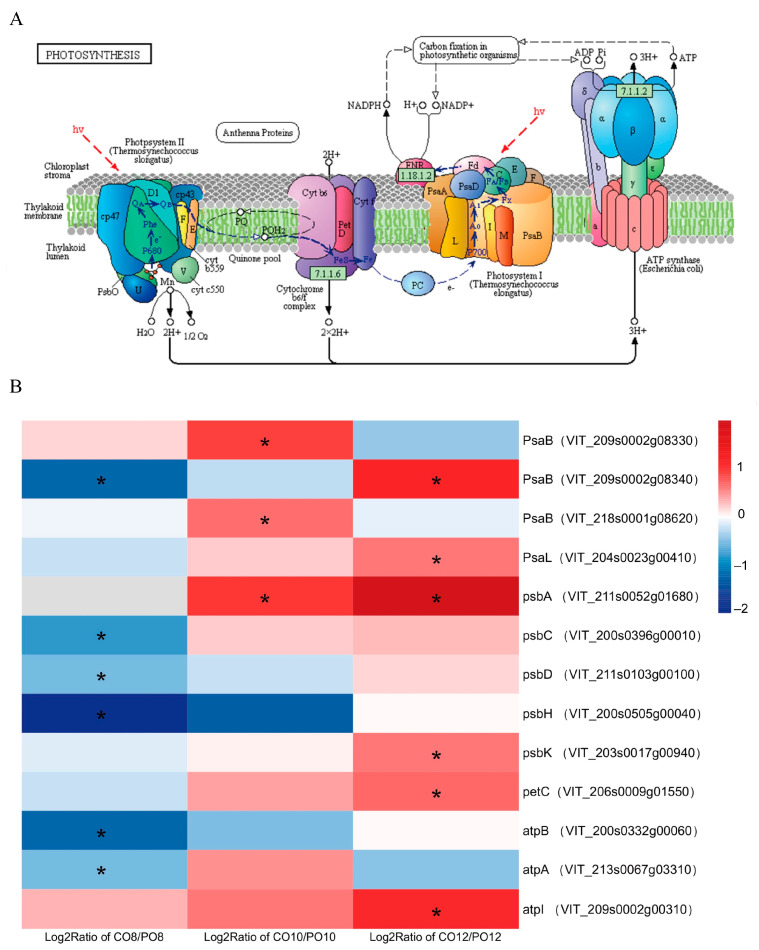
Differential expression analysis of photosystem II-related genes. (**A**) Schematic diagram of the light reaction of photosynthesis process. (**B**) Differentially expressed photosystem II-related genes in grape leaves under Coverlys TF150^®^ and PO film. Note: * indicates significant differences (*p* < 0.05).

**Figure 7 ijms-24-16659-f007:**
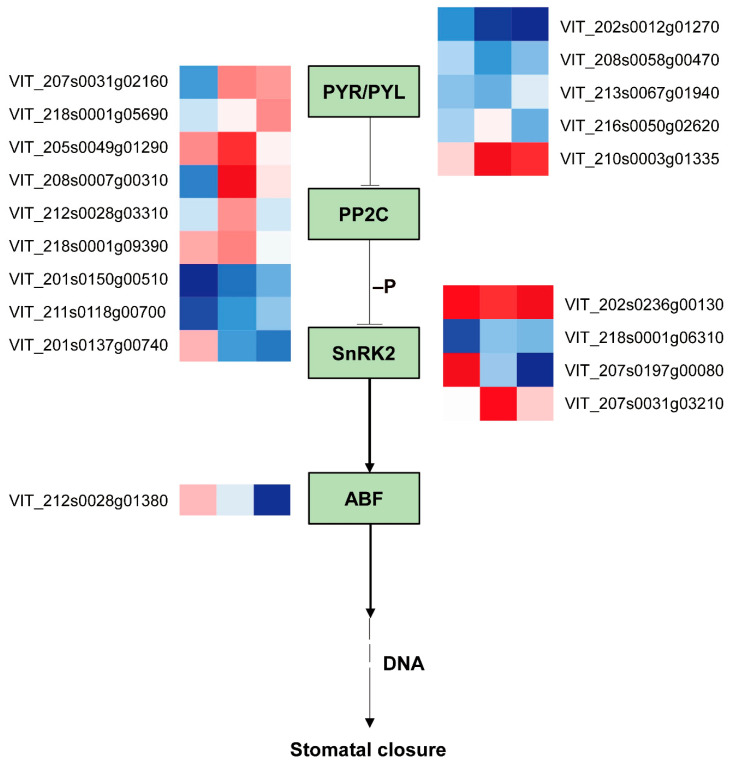
Differentially expressed genes in the ABA signaling pathway of grape leaves under Coverlys TF150^®^ and PO film. Note: PYR/PYL, abscisic acid receptor of PYR/PYL family; PP2C, protein phosphatase 2C; SnRK2, serine/threonine-protein kinase; ABF, ABA-responsive element binding factor.

**Figure 8 ijms-24-16659-f008:**
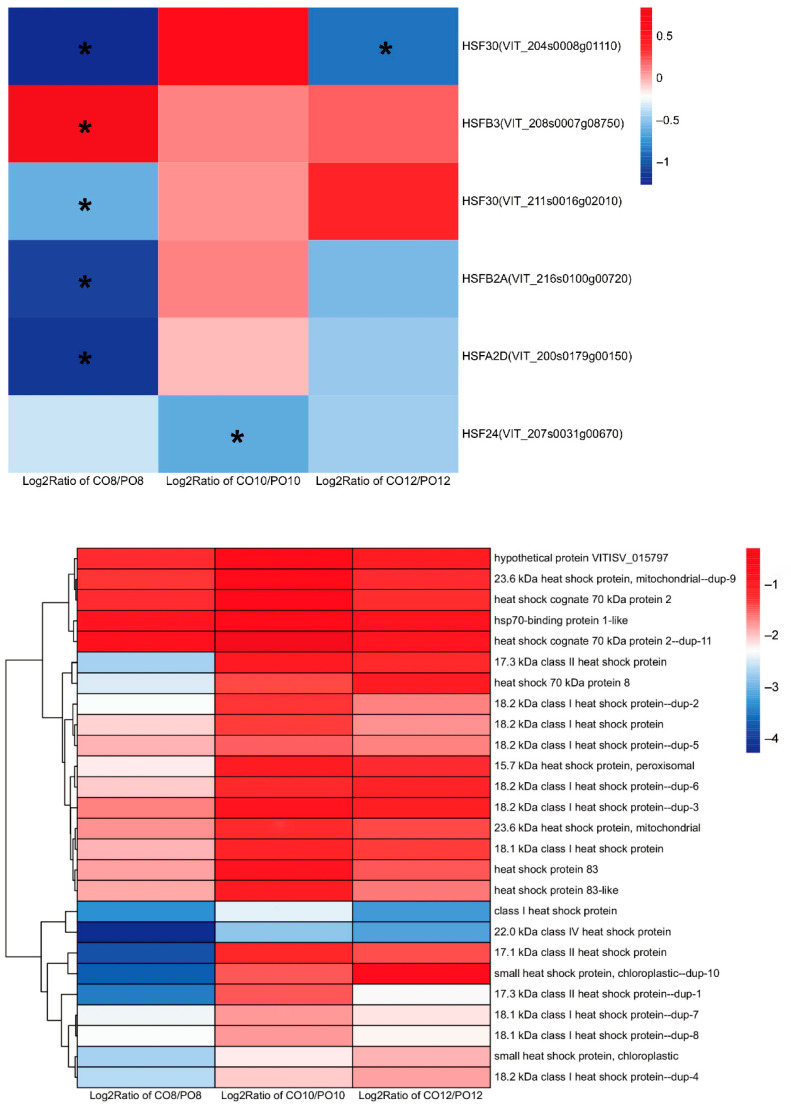
Differential expression of the heat shock protein and heat shock protein transcription factor genes in grape leaves under Coverlys TF150^®^ and PO film. Note: * indicates significant differences (*p* < 0.05).

**Table 1 ijms-24-16659-t001:** Effect of Coverlys TF150^®^ on grape leaf structure.

Treatment	Coverlys TF150^®^	PO
Upper epidermal thickness (μm)	26.88 ± 2.41 a	24.09 ± 2.52 a
Lower epidermal thickness (μm)	19.46 ± 1.66 a	22.25 ± 2.53 a
Palisade tissue thickness (μm)	101.32 ± 3.92 a	87.89 ± 5.08 b
Spongy tissue thickness (μm)	84.45 ± 10.22 a	78.76 ± 4.85 a

Note: Different letters indicate significant differences between treatments at *p* < 0.05.

## Data Availability

The original data for this present study are available from the corresponding authors.

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
