# Peer review of "Effects of Coverlys TF150® on the Photosynthetic Characteristics of Grape"

_ijms, 2023, doi:10.3390/ijms242316659_

Round 1

Reviewer 1 Report

Comments and Suggestions for Authors

This MS presents an extensive study assessing how a recently developed greenhouse covering affects microclimate, light dynamics, and associated physiological functioning of a Chinese grape cultivar compared to the most currently used covering.  Overall, the MS is well-written, the methodology and statistical analyses are solid, the results are clear and the authors do a good job of discussing their relevance.  Overall, this is a solid assessment of the effectiveness of the new cover.  I cannot, however, recommend the MS for publication in IJMS for the simple reason that the results do not provide much that is novel to the field of plant physiology or plant molecular biology.  There are other basic and applied journals specializing in just the kind of research presented here, such as HortScience, Agronomy, Environmental and Experimental Botany, Photosynthetica, etc. that would be a more appropriate outlet for this kind of study, and have the readership who would want to know about the effects of new coverings on plant physiological processes.

One minor point: the coverings induce changes in the microenvironmental conditions, which are their direct effect, while the associated plant responses track those.  As such, the coverings don't display a response, which the authors continually assert throughout the MS.  For example, lines 151-152, reads as though the different coverings had differing photosynthetic rates.  The coverings did not, the plants under them did.  I am fully aware of the authors' intent here, but as a basic courtesy to their readers, they should avoid this kind of mistake.

Comments on the Quality of English Language

There are the occasional misspellings such as "climat" for "climate" peppered throughout the MS that will require careful line-editing to address.

Author Response

Review Report (Round 1)

1.One minor point: the coverings induce changes in the microenvironmental conditions, which are their direct effect, while the associated plant responses track those. As such, the coverings don't display a response, which the authors continually assert throughout the MS. For example, lines 151-152, reads as though the different coverings had differing photosynthetic rates. The coverings did not, the plants under them did. I am fully aware of the authors' intent here, but as a basic courtesy to their readers, they should avoid this kind of mistake.

Response: Thank you for your comment. Careful and detailed modifications have been made.

  1. There are the occasional misspellings such as "climat" for "climate" peppered throughout the MS that will require careful line-editing to address.

Response: It has been modified in the text

Reviewer 2 Report

Comments and Suggestions for Authors

INTERESTING SUBJECT, RELATIVELY WELL CONDUCTED MEASUREMENTS, THE METHODOLOGY IS MISSING AN EXACT DESCRIPTION OF THE MATERIALS, THEIR SAMPLING, THE NUMBER OF REPLICATES, I WOULD ALSO WELCOME THEIR COMPARISON.

THE RESULTS ARE PRESENTED IN THE USUAL WAY FOR THE GIVEN FIELD AND USED METHODS.

THE DISCUSSION COULD BE MORE COMPREHENSIVE, THE APPLICABILITY OF THE RESULTS IN PRACTICE CAN BE APPRECIATED

SOME MINOR CORRECTIONS ARE REQUIRED.

Comments on the Quality of English Language

MINOR REVISIONS NEEDED

Author Response

Dear reviewers

Thank you for your letter . We were pleased to know that our work was rated as potentially acceptable for publication in Journal, subject to adequate revision. We thank the reviewers for the time and effort that they have put into reviewing the previous version of the manuscript. Their suggestions have enabled us to improve our work. Based on the instructions provided in your letter, we uploaded the file of the revised manuscript. Accordingly, we have uploaded a copy of the original manuscript with all the changes highlighted by using the track changes mode in MS Word.

Appended to this letter is our point-by-point response to the comments raised by the reviewers. The comments are reproduced and our responses are given directly afterward in a different color (red).

We would like also to thank you for allowing us to resubmit a revised copy of the manuscript.

We hope that the revised manuscript is accepted for publication in IJMS.

Round 2

Reviewer 1 Report

Comments and Suggestions for Authors

The previous MS had only minor editorial changes that needed to be addressed, and the authors have done so.  Since they have done so, the MS is acceptable for publication.  My main objection, which the authors have not addressed, is if IJMS is the most appropriate journal for this study.  I still feel that it is not, but will leave that decision up to the editors.